# OpenReview forum: "Tool-use Refiner: A Lightweight Plug-and-Play Module for Enhancing LLM Tool-Use"
_ICLR.cc/2026/Conference — ICLR 2026 Conference Withdrawn Submission_

### Official Review · Reviewer_2shu · 2025-10-27

**Soundness:** 2
**Presentation:** 3
**Contribution:** 2
**Rating:** 2
**Confidence:** 3

**Summary:**

The paper proposes a small post-processing model, the Tool-use Refiner, that takes a tool-integrated reasoning from the upstream LLM’s output, and outputs a corrected tool invocation. The Refiner is trained with DAPO and optimized using rule-based rewards that separately score format, tool names, parameters, and sequencing

**Strengths:**

The paper use RL to train a lightweight refiner to improve the performance of LLM tool calls, and show the performance gain across a number of downstream tasks.

The paper provides nice figures which help people easily interpret the core idea of the paper.

**Weaknesses:**

1. The authors stated that their appraoch is lightweight, however, compared to directly finetune the model, the approach needs extra parameters and training. The improvement based on this refinement is not seen greatly surpassing the baseline finetunning appraoch, so it's real world application is limited.

2. The approach is not really rooted in solving the tool calling problem of LLMs. This 'refiner' based approach is applicable to any type of errors an LLM made, and is not rooted in any specific observations on 'why LLMs can fail on tool calls'.

3. The author did not explore an end-to-end appraoch that roots in the mechanism of LLM inference, so providing less insights to future studies in the area.

**Questions:**

What's the optimal design of the proposed refiner appraoch? i.e How many parameters and training data is needed for getting a good refiner which can improve the original model of what size in what percentage ? The effectiveness of the approach is not explored well in the current experiments, and a number of ablation study is needed to inform the community with the effectiveness of the appraoch.

---

### Official Review · Reviewer_YcMw · 2025-10-31

**Soundness:** 2
**Presentation:** 2
**Contribution:** 2
**Rating:** 4
**Confidence:** 4

**Summary:**

This paper introduces a lightweight, plug-and-play module called the Tool-use Refiner, designed to improve tool-integrated reasoning in LLMs. Instead of fine-tuning or retraining the upstream LLM, the authors propose adding a small auxiliary model (Qwen3-1.7B) that takes the original model's tool invocation output and corrects its formatting, parameter values, tool ordering, and other semantic errors. The Refiner is trained using the DAPO reinforcement learning algorithm with a rule-based reward function tailored for tool-calling accuracy. Experiments across several benchmarks show that this refinement stage significantly improves the performance of smaller upstream models, and provides consistent gains across different LLM types and sizes.

**Strengths:**

- The method is straightforward and achieves clear performance improvements on smaller upstream models.
The proposed pipeline adds a secondary model to refine tool outputs, which is easy to understand and implement, yet leads to tangible improvements in task accuracy. The results on benchmarks demonstrate meaningful gains for models in the 1.7B–3B scale, which highlights the practical utility of the approach for resource-constrained settings.

- The paper is generally clear in its problem formulation and experimental setup.
The motivation is well-grounded in the common observation that LLMs frequently mis-handle tool usage even with structured prompts. The authors clearly describe the design of the refiner, the reward functions, and the evaluation protocol, making it relatively easy to reproduce and follow the logic of the pipeline.

**Weaknesses:**

- It is unclear whether the proposed gains persist for larger models that better handle tool invocation.
The improvements are most pronounced for weaker models, and much smaller for stronger baselines. This raises the question of whether the approach is necessary or beneficial at all in scenarios where powerful LLMs are already used.

- The method introduces an extra inference stage, which may become a bottleneck in practice.
Although the Refiner is small, the approach requires invoking two models sequentially. This post-hoc fixing mechanism increases latency and complexity at inference time. Moreover, it is not scalable to assume that similar patching modules will be needed for every new downstream capability where the base model exhibits weakness. The paradigm is generally not preferred in LLM development.

- The technical contribution is largely an application of the DAPO algorithm to a specific downstream setting.
The core approach is to train a small model with RL to refine another model’s output, which does not introduce significant algorithmic or engineering innovation. The novelty mainly lies in its application to tool-use correction, and relies heavily on prior work in both refinement-based model design and DAPO training.

**Questions:**

see above

---

### Official Review · Reviewer_9RXN · 2025-11-02

**Soundness:** 2
**Presentation:** 3
**Contribution:** 3
**Rating:** 6
**Confidence:** 3

**Summary:**

This paper presents a novel and practical approach to a significant problem in Large Language Models (LLMs): frequent errors in tool invocations (e.g., incorrect parameters, malformed formats). Instead of the computationally expensive process of fine-tuning large upstream LLMs, the authors propose a lightweight, plug-and-play "Tool-use Refiner" model. This smaller model acts as a post-processing module, taking the upstream LLM's potentially flawed tool-call output and the user's original query to generate a corrected and enhanced invocation, trained using a sophisticated Reinforcement Learning (RL) algorithm, DAPO.

**Strengths:**

1. Innovative and Efficient Architecture: 1) The core idea of using a small, specialized model (Qwen3-1.7B) to correct the outputs of larger, frozen models is both clever and resource-efficient. It avoids the high cost of directly training or fine-tuning large LLMs. 2) The plug-and-play design ensures broad compatibility and easy integration with various upstream LLMs, making it a versatile solution.

2. Comprehensive and Well-Designed Training Methodology: 1) The training data construction is meticulous, systematically generating a diverse set of error types (format, semantic, sequential) using multiple LLMs to simulate real-world failure modes. 2) The rule-based reward function is multi-faceted and well-structured, separately evaluating format correctness, tool name accuracy, parameter fidelity, and invocation order. The "regression penalty" is a particularly smart addition to prevent the Refiner from degrading already-correct inputs.

3. Thorough and Convincing Experimental Validation: 1) The paper provides an extensive evaluation across multiple, diverse benchmarks (API-Bank, NESTFUL, Musique, Bamboogle), covering general-purpose, sequential, and search tool invocations. 2) The results consistently show performance improvements across upstream LLMs of various scales, effectively demonstrating the general applicability of the method.

**Weaknesses:**

1. Inherent Complexity in Training: 1) While avoiding large LLM training, the proposed pipeline introduces its own complexity. The process of curating the training data using multiple LLMs and the need for a carefully tuned, multi-component reward function are non-trivial and require significant expertise. 2) RL training for language generation is notoriously unstable and sensitive to hyperparameters, which could pose reproducibility and deployment challenges despite the use of the modern DAPO algorithm.

2. Constraints on Scope and Applicability: 1) The approach is primarily focused on the syntactic and semantic correctness of the tool call itself, not on addressing higher-level reasoning failures in the upstream model's plan. 2) In a production environment, the method may face challenges with adaptability, as any changes to the available tool set or their APIs would likely require re-training or fine-tuning the Refiner.

**Questions:**

1. this two-stage tool-use is similar to a first-stage cot (may be a wrong tool-calling) and sencond-stage tool use generation. previous methods like [1,2] have explore this, could you compare them?
2. What was the rationale for selecting the DAPO algorithm over other native tool-use RL methods like [3,4]? Were any comparative experiments conducted with other RL algorithms to validate this choice?

[1] Tptu: Task planning and tool usage of large language model-based ai agents
[2] Tptu-v2: Boosting task planning and tool usage of large language model-based agents in real-world systems
[3] Agentic reinforced policy optimization
[4] Agentic Entropy-Balanced Policy Optimization

---

### Official Review · Reviewer_kMoh · 2025-11-02

**Soundness:** 3
**Presentation:** 3
**Contribution:** 2
**Rating:** 2
**Confidence:** 4

**Summary:**

This paper tackles the problems for frequent tool invocation errors in LLMs such as incorrect parameters, invalid parameters, or improper tool-call ordering. The authors argue that while training with SFT/RL can mitigate these issues, they are computationally expensive, especially for larger models. The authors approached this by proposing a “refiner”, a small-scale model that refines the tool calls in a post-hoc fashion, which takes the user query and the potentially flawed tool invocation from a frozen upstream LLM and generates a refined version. The refiner model is trained with a policy optimization algorithm (DAPO), guided by several rule-based reward functions that the authors designed. Experiments on several benchmarks show that the proposed model can improve general-purpose single and sequential tool invocations, and refine search tool invocations.

**Strengths:**

The proposed refiner approach is modular and practical, offering an alternative to the costly full-model fine-tuning for a real-world problem. The paper is well-written and easy to follow.

**Weaknesses:**

There are critical flaws in the experimental settings. See comments below.  A fundamental weakness is the practical cost-benefit tradeoff. The reported results show that the refiner provides the largest performance gains for smaller models where the relative inference cost is the highest; in contrast, it offers less gains for larger models, where the additive cost of a 1.7B refiner would be most justifiable.

For the main results in Table 1, the baselines (SFT, PPO, etc) are trained on a 4k dataset (400 for SFT; Appendix E) sampled from several benchmarks, while the proposed refiner is trained on a different (constructed) dataset (Sec 3.2), sourced from different benchmarks. This renders the comparison invalid.

While the authors argue to use RL to train the refiner, they never provided an empirical comparison against a simple SFT-trained refiner with the constructed training data. The mapping of flawed output + user query (potentially +CoT traces) to refined output is a supervised task, and should be compared against to justify using the more complex RL training design.

For the first evaluation setting of general-purpose invocation correction, which is the most critical set of experiments to support the paper’s central claim, it was conducted on only a single, relatively older and smaller benchmark. This is insufficient, especially when benchmarks like BFCL are very commonly used in literature.

In the experiments, the authors did not test on SOTA fine-tuned models such as ToolACE/Hammer, nor on SOTA closed models such as GPT. Including these results would strengthen the claims.

Furthermore, the experiments fail to test for cross-model generalization. The refiner was trained on errors generated by the same model families (Llama, Qwen) it was evaluated on, showing it can learn to fix a model’s own error patterns, but not that it can generalize to unseen models. Including these results would strengthen the plug-and-play claim.

For sequential or multi-turn tool calls, the proposed method’s design is strictly post-hoc, correcting the entire tool sequence after it has been generated. It would be good to have an experiment for comparing this approach with applying a step-wise refinement, to potentially correct flawed reasoning before the sequence is unsalvageable, as the authors noted that CoT reasoning traces are critical to the performance of the refiner.

**Questions:**

The authors can address the weaknesses outlined in the section above.

---

### Note · Authors · 2025-11-30

I have read and agree with the venue's withdrawal policy on behalf of myself and my co-authors.